# Cover cropping impacts on soil water and carbon in dryland cropping system

**Hanlu Zhang** [1,2]*, **Afshin Ghahramani**[1,2]*, **Aram Ali**[1,3], **Andrew Erbacher**[4]

**1** Centre for Sustainable Agricultural Systems, Institute for Life Sciences and the Environment, University of Southern Queensland, Toowoomba, Qld, Australia, **2** Cooperative Research Centre for High Performance Soils, Callaghan, NSW, Australia, **3** Soil and Water Science Department, College of Agricultural Engineering Sciences, Salahaddin University-Erbil, Erbil, Kurdistan Region, Iraq, **4** Department of Agriculture and Fisheries, Queensland Government, Goondiwindi, Qld, Australia

* hanlu.zhang@usq.edu.au (HZ); afshin.ghahramani@usq.edu.au (AG)

## Abstract

Incorporating cover crops into the rotation is a practice applied across many parts of the globe to enhance soil biological activities. In dryland farming, where crop production is highly dependent on rainfall and soil water storage, cover cropping can affect soil water, yet its effects on soil hydrological and biological health require further investigation. The objective of this study was to evaluate the effect of different timing of summer sorghum cover crop termination on soil water, total and labile organic carbon, arbuscular mycorrhizal fungi and their mediating effects on wheat yield. Through on-farm trial, soil characteristics along with wheat biomass, yield and grain quality were monitored. In comparison with the control (fallow), the early terminated cover crop was the most effective at retaining greater soil water at wheat sowing by 1~4% in 0–45cm soil profile. An increase in water use efficiency, yield and grain protein by 10%, 12% and 5% was observed under early termination. Under late terminated summer cover crop, there was 7% soil water depletion at wheat planting which resulted in 61% decline in yield. However, late-terminated cover crop achieved the greatest gain in soil total and particulate organic carbon by 17% and 72% and arbuscular mycorrhizal fungal Group A and B concentration by 356% and 251%. Summer cover crop incorporation resulted in a rapid gain in labile organic carbon, which constituted hotspots for arbuscular mycorrhizal fungi growth, conversely, fungal activities increased labile organic carbon availability. The combined effect of increased soil water at sowing and over the growing season, organic carbon, and microbial activities contributed to greater yield. The findings suggest that summer cover cropping with timely termination can have implications in managing soil water at sowing time and enhancing soil water storage during the season, soil carbon, and facilitating microbial activities while enhancing productivity in the dryland cropping system.

## 1. Introduction

Soil water is often a limiting factor in Australian crop production regions particularly cropping systems in the states of New South Wales (NSW) and Queensland (Qld), where stored moisture in the soil profile at sowing time is critical for seedling emergence, crop establishment and

**Data Availability Statement:** All relevant data are within the paper.

**Funding:** "This work has been supported by the Cooperative Research Centre for High Performance Soils (Australian Government's Cooperative

Research Centre program), along with support from both the Queensland Government Department of Agriculture and Fisheries and the University of Southern Queensland through the Broadacre Cropping Initiative."

**Competing interests:** The authors have declared that no competing interests exist.

yield [1–5]. However, this importance can be affected by management that changes soil biological activities and nutrient availabilities [6, 7]. Low soil water availability can impair nutrient availability by affecting nutrient concentration in soil solution and the rate of nutrient transport to the root, affecting plant growth and yield [8]. On the other hand, soil water availability can also affect microbial growth, microbial activities and their physical and chemical processes that mobilise organic matter via root exudates [9]. There is a range of cropping practices such as early sowing, crop rotations, stubble retention, minimum tillage or no-till and weed control to improve water use efficiency by enhancing capture and preservation of rainfall [10, 11]. Among these, the use of cover crops has been adopted across many parts of the globe to manage both soil water and nutrients but different effects on soil water have been reported for these practices [12–16]. Integrating cover crops into a crop-fallow system can be a method to replace or shorten the fallow duration, which allows longer duration of soil surface coverage before planting the cash crop [17]. Fallow replacement with a cover crop can affect soil water dynamics by regulating soil water evaporation, runoff and drainage [18] and affecting microbial community structure and consequently biological activities [19]. The use of cover crops in cropping systems provides benefits of modifying the soil environment and enhancing soil physical properties through its effect on root-soil interactions, but their impacts on soil physical and biological characteristics were reported to vary in different environments and cropping systems [20–24]. Long-term cover crop practices can lead to changes in soil hydraulic properties, such as soil bulk density, aggregate stability, soil water retention, infiltration, saturated hydraulic conductivity and pore-size distributions across the soil profile [17]. Nevertheless, the magnitude of changes can be highly site and management-dependent [25–27].

Cover crops can be the source of plant residues from above-ground biomass and root biomass that contributes to the organic matter pool after decomposition, which improves soil hydrology that can potentially increase soil water storage and plant available water [28, 29]. Studies showed that long-term cover cropping contributed to a better-developed soil structure by improving soil pore size distribution and soil hydraulic properties, such as soil water conductivity and retention at the plot scale [30]. However, the impact of cover crop practice on soil water can vary across years and regions that are affected by climate variability and soil types [27, 31, 32]. Under dryland conditions cover crops are more likely to compete for soil water and nutrient resources with cash crops [15, 31]. Therefore appropriate termination time becomes crucial for cover crop management to avoid the competition and reduce soil water loss from evapotranspiration, particularly in water-limited regions where soil nutrients and water-use efficiency are often low [33–35]. Cover crop management has also been reported as an effective practice to improve soil chemical and biological characteristics, such as enhancing nitrogen recycling via reduced nitrate leaching risks, increasing soil organic carbon (OC), and microbial biomass and activities [36–39]. In general, the main drivers of the net change of soil total carbon are the organic matter from plant residues, soil biota metabolisms, and organic amendments which the first two can be supplied or sustained by cover cropping [40, 41]. Cover crops affect soil organic matter (SOM) and different forms of carbon in soil, i.e., total carbon, organic carbon and different forms of active or labile organic carbon (LOC) that are often known as the particulate organic carbon (POC) and permanganate oxidizable carbon (POXC or MnoxC). Soil OC is the carbon component of SOM, which can be 58–60% of SOM [42]. Soil POC and POXC are the small and active fractions of the soil TOC pool, and their lability (undergoing breakdowns) has been reported to have a relationship with the biomass of living organisms [43, 44]. LOC fraction as an active soil OC component is mostly derived from fresh organic materials and often correlated with the dynamic of SOM, and is highly sensitive to soil management [45, 46]. Soil with improved SOM management is likely to have higher productivity due to increased LOC [44]. In a semi-arid dryland cropping system, planting

cover crops to replace fallow can vitalise soil aggregation through direct addition of SOM, promoting microbial activities, binding of soil particles by roots or fungal hyphae, and aggravation of wet-dry cycles due to evapotranspiration [47].

Improved SOM or soil OC can lead to changes in soil physical characteristics and potentially soil water characteristics [48]. However, the response of plant available water capacity (PAWC) and soil water retention to variation of SOM or soil OC were reported to differ as SOM varies with soil texture [49, 50]. For example, an increase in SOM can decrease evaporation and suppress infiltration and hence increasing soil water retention during cover crop growth stages and improving water use efficiency [51]. On the other hand, an increase in SOM may decrease soil water retention for heavy clay soils [49].

In dryland environments, soil microbial communities are considered as another crucial component, which play an important role in coordinating water and nutrient inputs and outputs, which consequently can affect nutrient cycles and hydrological cycles [52]. In particular, arbuscular mycorrhizal fungi (AMF) play a critical role in soil-plant interactions such as stimulating residue decomposition, facilitating plant nutrient and water uptake, and facilitating soil carbon cycling [53, 54]. The interaction between soil water and AMF is often associated with host plants. Soil water availability has a direct impact on plant root lifespan and turnover, consequently, affecting AMF community composition and symbiosis [55, 56]. AMF regulate soil water content through hyphal colonisation and glomalin-related soil proteins (GRSP), which promotes soil aggregation and soil physical structural stability [57, 58]. Meanwhile, GRSP consists of 30–40% carbon and its related compounds were found to be beneficial in improving soil water holding capacity and hydraulic conductivity, and subsequently positively correlated with plant available water content [59, 60]. The presence and decomposition of cover crops can affect the characteristics of the microbial communities, such as variation and structure. [61]. Cover cropping, particularly with no-till practice can not only enhance root colonization from AMF and possibly shifting AMF community structure during cover cropping season [62], but also enhance early mycorrhizal colonization of the following crop and assist the success of seedling establishment [63, 64]. Cover cropping can be a potential way to improve available carbon, AMF colonization and population nutrient accessibility [65, 66], and potentially facilitate the symbiotic relationship between AMF and crop roots for exchanging water and nutrients for carbon [67].

Overall, cover crop incorporation provides a range of environmental benefits, such as improved soil physical and biological properties, improved soil water and nutrient availability, and reduced soil carbon decline rate [22, 68–71]. Facing the uncertainty of climate variability and increasing food demand, strategic deployment of cover crop practices can be of support for maintaining the function and resilience of agroecosystems [72]. The effectiveness of cover cropping has been reported to vary across many parts of the globe and limited previous works exist on how cover crops affect soil and system productivity within a cropping system in Australia, in particular, the state of Qld [73, 74]. The objectives of this study were to evaluate the effect of summer cover crop practices on 1) soil OC (i.e., TOC, POC and POXC) and soil AMF DNA sequence concentrations at termination time of summer cover crop; 2) soil water across the soil profile (i.e. 0–150 cm) over the growing season and at the sowing time of the following cash crop; and 3) investigate the dependencies between soil OC and soil water at planting, wheat biomass, yield and grain quality.

To address the above questions and explore the effects of cover crop on soil health and cash crop yield production over growing seasons in the rainfed agricultural system, on-farm trial was conducted to monitor soil water across the soil profile, soil OC and AMF DNA concentrations, wheat biomass, yield and grain quality in a wheat cropping system planted after a summer sorghum cover crop. Different terminations of summer cover crops were applied to

manage soil water and carbon. This paper aimed to improve our understanding of the plant-soil-water relations in the rainfed cover cropping system in the eastern region of the wheat belt, where crop production is highly dependent on soil water at planting time and to investigate whether cover management can influence soil water characteristics and soil carbon accumulation.

## 2. Material and methods

### 2.1 Study site

The field experiments were carried out on a farm located north of Goondiwindi region, in the Southwest of Queensland State, Australia (Fig 1). The average annual rainfall of the region is approximately 486 mm (summer dominant), with monthly mean temperature ranging from 11.5 to 27.0˚C [75]. The study site represents a crop growing region in Australia, as it is within the northern part of the grain belt region, where cereal crops (such as wheat, oats, barley, sorghum and maize) are grown in an extremely variable climate [76]. Water supply (rainfall) and storage (soil water storage) are the major limiting factors for dryland grain production in the region [77]. In 2015/2016, the Goondiwindi region's cereal production was the second largest commodity of agricultural production in the region, which accounted for 28.2% of Goondiwindi Regional Council's total agricultural output in value terms (AU$530 million) [78].

The study site had been managed under grain cropping-fallow systems by the landowner. The site experienced long fallow in 2019 due to the drought condition which could potentially alter soil microbial composition and activities (e.g., microbial decomposition of soil organic matter) with a corresponding consequence in soil carbon and nitrogen balance [79]. In 2020, winter wheat was planted in May and harvested in October, with wheat stubble left standing in the field.

Experimental trials are part of the project the Broadacre Cropping Initiative (BACI) supported by the Queensland Government (Department of Agriculture and Fisheries) and the University of Southern Queensland. All the approvals have been obtained for conducting this research and information such as property name and coordinates cannot be disclosed for confidentiality reasons. Our trials were conducted in 2021 with extended summer rainfall (Jan–May) 2% below the 1990–2020 average, which rainfall distribution over three months was greater than average only at the start of summer (sowing time of summer cover crop) but became significantly lower than average three months before planting the winter wheat in late May (Fig 1). Thus, the examined year is a good example of a seasonal condition in an area where winter crop yield is highly dependent on soil water storage [80]. The soil is classified as a vertosol [81] with a high clay content that ranged between 40 and 60%. Vertosols have shrinking and swelling characteristics in response to the changing soil water content which is related to the changes in interparticle and intraparticle porosity [82]. Fig 1(C) shows examples of the cracking soil surface at the trial site.

### 2.2 Trial design

Field trials were conducted during the 2021 summer and winter seasons. The trial design for the summer cover crop season was a randomised complete block design [83] under the uniform paddock condition with five replicates per treatment, including fallow treatment as control. Cover crop plots were terminated by spraying at three different stages early, mid and late stages (Table 1). For the comparison, the control plots remained as fallow as is a common practice during the summer season before planting the cash crop, Therefore, a total number of 20 zero-tilled plots were used for the summer cover crop trial with 5 replicates for each treatment. For the winter season, the trial design was based on a split-plot design [84] which equally

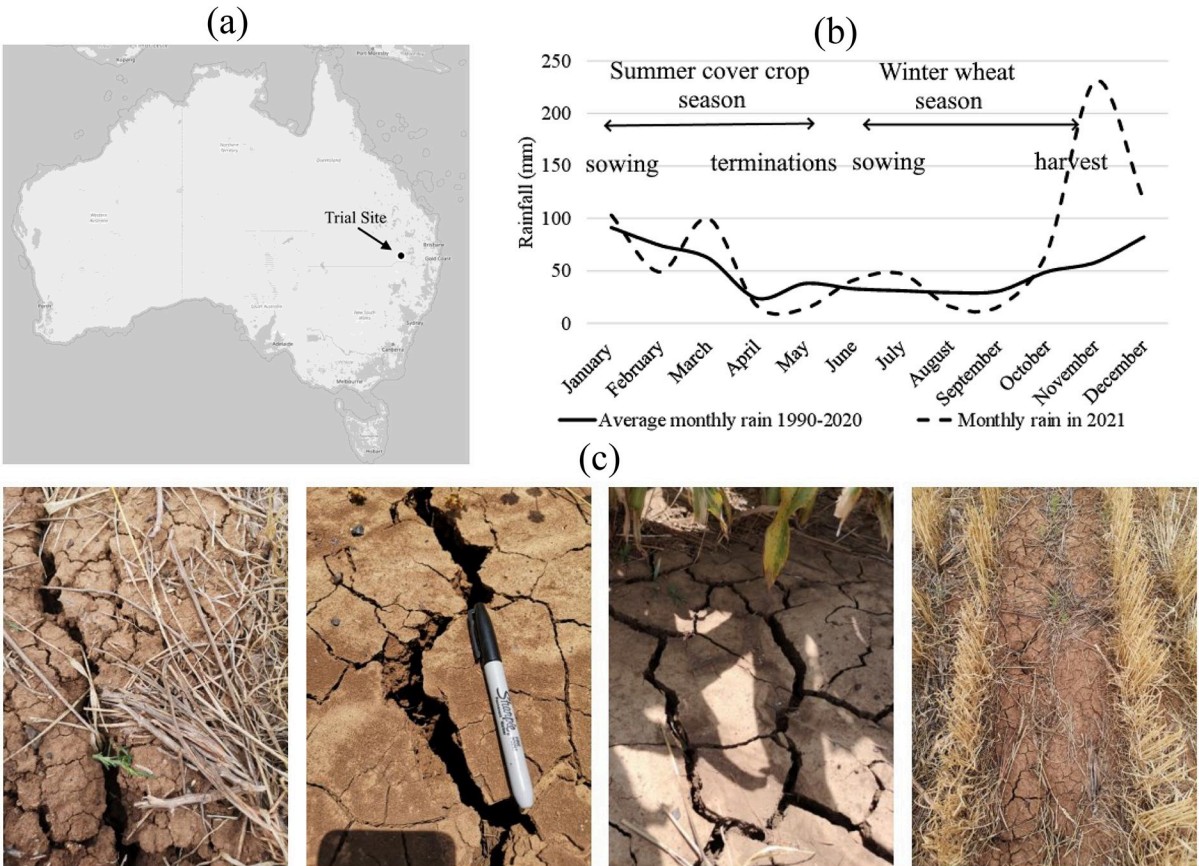

**Fig 1.** Case study site (a) approximate location of trial site, (b) monthly average rainfall of the season in comparison with the long-term average, (c) soil surface at the trial site (Photos taken by Hanlu Zhang in 2021). Cracking soil surface is a characteristic of the vertosols of the study site.

divided each plot into 2 sub-plots. The treatments for the paired subplots were wheat and fallow (as control). Therefore, a total number 20 sub-plots were planted with wheat and another 20 sub-plots were left as fallow.

## 2.3 Sampling

Soil sampling from replicate plots of each treatment conducted during the summer cover crop trial and winter wheat trial in 2021. The sampling included collecting soil cores for measuring soil physical characteristics such as bulk density, particle size distribution, soil water content across the profile, and surface soil samples for measuring soil OC and its active fractions, and the AMF DNA sequence concentrations (Table 2). The related soil attributes were collected to evaluate the effect of cover crop and its termination management on soil water, soil OC and labile fractions, and AMF DNA sequence concentrations over the growing seasons as presented in Table 2. These relevant soil attributes were collected for analysis at a system level.

**2.3.1 Soil water.** Soil profile samples were collected for measurement of bulk density, particle size distribution, and gravimetric and volumetric soil water content. Intact soil cores (4.3 cm diameter by 5.0 cm) up to 1.5 m of soil profile were collected from each treatment plot using a hydraulic soil sampling rig. Soil cores were cut into 10.0 cm height sections and placed into PVC columns for safe storage and transportation. The soil cores were processed in the

**Table 1. Components of the farm system at the trial site.** Weather records are for the growing season (Jan-Nov 2021).

| Weather | Records |
| --- | --- |
| Rainfall (mm) | 542.8 |
| Mean max temperature (˚C) | 25.7 |
| Mean min temperature (˚C) | 12.3 |
| **Summer cover crop trial** | **Trial information** |
| Plot size (m) | 18 x 6 |
| Crop cultivar | sorghum (MR Bazley) |
| Summer control (SC) | Fallow (sprayed and no weeds) |
| Planting date | 15/01/2021 |
| Sowing depth (cm) | 3 |
| Row space (cm) | 25 |
| Early termination (ET) | 2/03/2021 (flag leaf emergence) |
| Mid termination (MT) | 30/03/2021 (soft dough) |
| Late termination (LT) | 21/04/2021 (hard dough) |
| **winter wheat trial** | **Trial information** |
| Crop cultivar | wheat (Suntop) |
| Winter control (WC) | Fallow (sprayed and no weeds) |
| Sowing depth (cm) | 30 |
| Row space (cm) | 25 |
| Planting date | 28/05/2021 |
| Harvest date | 27/10/2021 |
| **Chemical used** | **Application rate** |
| Herbicide application | Roundup UltraMax 0.2 ml/ha applied at each termination (Active ingredient 570g/L Glyphosate) |
| Fertiliser application | Starter Z 25kg/ha on 15/01/2021 and 40kg/ha on 28/05/2021 (Mono ammonium phosphate plus zinc, containing 10% N, 22% P, 2% S and 1% Zn) |

laboratory for initial field weight and oven-dried weight through oven dry at 105˚C for a minimum of 48 hours to determine the bulk density and gravimetric soil water content. Due to the cracking clay characteristics of the vertosol soils, it is challenging to accurately measure soil water content among various proximal sensors [85]. For this study, neutron moisture meters (NMM) were used to regularly measure point-source soil water in the field. Soil water content measurement using NMM has a better representative value as the measurement sphere is up to a 15 cm radius around the emitted neutron source. In this way, soil cracks are less likely to affect the reading (Fig 2). Soil water monitoring using the NMM approach was based on the physical interaction of radioactive neutrons with hydrogen atoms, and it had better control of both bulk density and hydrogen atoms during the calibration process [86]. On the basis of the positive relationship between the relative neutron count rate and volumetric soil water, soil water content was estimated from the NMM readings (Fig 2). Prior to the planting of the summer cover crop, a total number of 20 aluminium NMM access tubes were installed at the centre of each plot, which allows taking neutron counts for each soil depth at 15cm, 35cm, 45cm, 55cm, 75cm, 105cm and 135cm. At the end of the summer trial, NMM access tubes were all removed for the preparation of winter wheat planting. 40 tubes were reinstalled after planting and resumed NMM reading measurements for all 40 sub-plots. NMM readings were taken regularly as part of soil water monitoring during the growing season. NMM readings were

**Table 2. Soil attribute tests performed in this study and the purpose of conducting those tests.**

| Attribute | Why measure? What does this represent? | Method reference |
|---|---|---|
| Bulk density (BD) | A physical characteristic of soil represents soil structure and compaction. Can be an indicator of soil health in response to changes in management. | Blake and Hartge 1986 [87] |
| Soil Particle Distribution | A physical characteristic of soil that drives water holding capacity and flux movement. | Gee and Bauder 1986 [88] |
| Gravimetric Soil Water (GSW) Content | It is the % soil water on a dry-mass basis; critical to plant growth, nutrient movement and microbial activity | Reynolds 1970 [89] |
| Volumetric Soil Water (VSW) Content | It is the ratio of soil water volume to the volume of soil, can be calculated from measured GSW multiplied by BD | |
| Total Organic Carbon (TOC) | Stored in soil organic matter; C component of organic compounds; An indicator of soil health and biology. | Sweeney and Rexroad 1987 [90]; Etheridge et al. 1998 [91] |
| Particulate Organic Carbon (POC) | Fresh or partially decomposed plant residue and animal matter with identifiable cell structure. Makes up 2–25% of total soil organic matter; labile OC pool [92] | Blair et al. 1995 [93] |
| Permanganate-oxidisable Carbon (MnoxC or POXC) | A sub-pool of labile soil OC is defined as the C that can be oxidized by potassium permanganate (KMnO4) [92] | Cambardella and Elliot 1992 [94] |
| Arbuscular Mycorrhizal Fungi (AMF) | Important microbial communities that regulate plant growth [58] and contribute to soil aggregate formation and stability [57] | Sanders et al. 1995 [95]; Senés-Guerrero and Schüßler 2016 [96] |

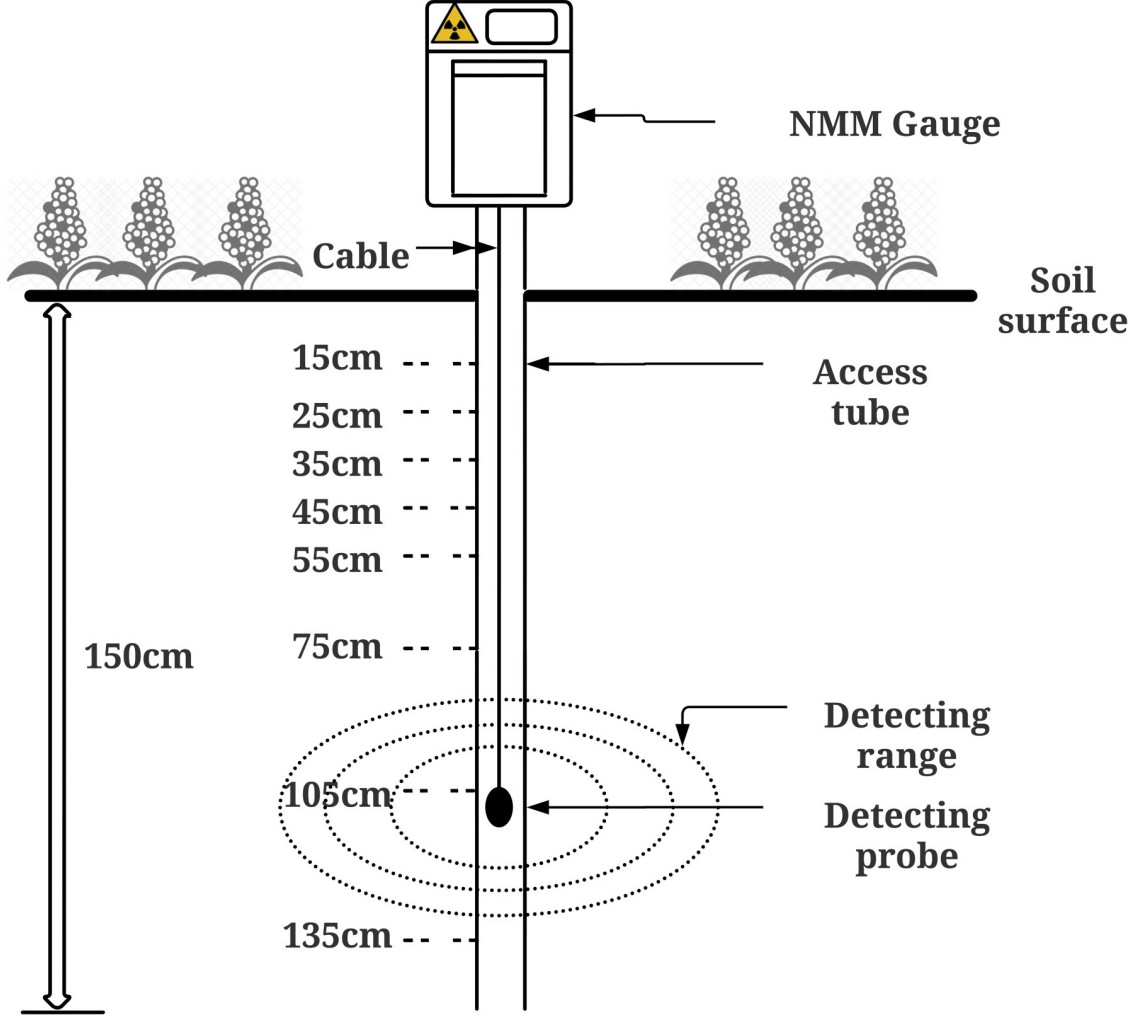

**Fig 2. Use of neutron moisture meter probe in the field.**

collected by a licenced technician at 2–3 weeks intervals. By using the dry weight of the soil cores (43 mm diameter by 500 mm length) collected from each depth at the time of neutron probe access tube installation, soil bulk density and gravimetric water content were measured, then converted to volumetric soil water content. The neutron probe was calibrated against gravimetric soil water content, soil texture and bulk density for each plot. Soil particle size distribution of the whole soil profile was measured for all plots.

**2.3.2 Soil carbon.** Top 10cm soil samples were collected (10 subsamples per plot) at the termination time of summer cover crop to monitor soil OC content and its labile fractions POC and POXC (Table 2) to be explored along with soil water content. At the end of the winter trial, soil samples were sampled for TOC and POC contents (10 subsamples for subplot). The soil samples were tested by commercial laboratories (Chemistry Centre, Department of Environment and Science, Queensland Government, Qld; the Environmental Analysis Laboratory, South Cross University, NSW), to determine total OC content [89, 90], POC [94] and POXC contents [93]. The samples were air-dried at 40°C and ground to pass through a 2 mm sieve before the test. The instrument used for soil total OC measurement was TruMac Carbon/Nitrogen Determinator (LECO Corporation, St. Joseph, MI, USA). The carbon content of the soil samples was determined by analysing the amount of carbon dioxide produced from the combustion of the sample at a high temperature based on the Dumas combustion method [97]. Measurement of soil POC content was measured by dispersing soil samples in Calgon (sodium hexametaphosphate) to extract soil fraction >50um, then processed for the carbon determination in LECO [94, 98]. Soil POXC measurement used $MnO_4^-$ solution ($KMnO_4$) to react with the soil sample and determined the POXC content based on the degree of oxidation [93].

**2.3.3 Soil arbuscular mycorrhizal fungi (AMF).** The soil samples in the top 10 cm were collected from each summer cover crop treatment and their corresponding control plot to test AMF DNA sequence concentrations different species groups (Table 3). At each plot, 10 subsamples per plot were collected to form one sample (approximately 500g/sample) that represents the whole plot's area. The samples were tested at a commercial lab i.e., the South Australian Research and Development Institute (SARDI) laboratory for DNA-based characterization and identification of AMF from different phylogeny taxa groups. In the laboratory, to identify the AMF DNA concentration of each functional group, AMF spores were extracted from the soil samples using sucrose centrifugation and flotation, followed by polymerase chain reaction (PCR) DNA extraction to perform molecular analysis [95, 96]. The AMF test measured the DNA sequence concentration in each sample and assigned to their phylogeny taxa using the maximum likelihood method based on near full length small ribosomal subunit (SSU) rRNA sequences [99]. The results exhibited the existence of groups A and B (Table 3), these two groups are from the genus of *Funneliformis* and *Claroideglomus* [99, 100]. The functional diversity of AMF such as the function of mycorrhizal symbiosis and its symbiotic efficiency is genotype dependent and can be complex to study the characteristics of species individually [101]. Therefore, for simplicity, the groups A and B were identified based on the DNA sequence, which was used for analysis in this paper, rather than the individual species in each group

**Table 3. AMF species groups and species in each group.**

| Indicator | Group | Species |
|---|---|---|
| Arbuscular mycorrhizal fungi (AMF) | Group A | *Funneliformis mosseae, Funneliformis constrictum, Funneliformis coronatum, Funneliformis geosporum, Funneliformis verruculosum, Funneliformis caledonium* and *Funneliformis fragilistratum* |
| | Group B | *Claroideglomus claroideum* and *Claroideoglomus etunicatum* |

(Table 3). AMF DNA sequences in soil and their variation under different summer cover crop treatments were observed to explore whether there were linkages to the changes (e.g., soil water).

**2.3.4 Crop.** During the summer cover crop trial, above-ground sorghum biomass was sampled at the time of each termination using a 0.25 m$^2$ quadrat and five random sampling within the plots. The samples were oven-dried at 70°C for 72 hours and then weighed to measure dry biomass weight. During the winter season, above-ground wheat biomass from each plot was collected at two different growth stages i.e., grain filling and early maturity phenology stages, and collected yield at the harvest. The biomass was oven-dried and weighed using the same procedures explained earlier. Grain samples were analysed by a commercial lab (Leslie Research Centre, Department of Agriculture and Fisheries, Queensland Government) to test grain quality i.e., grain protein, and wheat screenings. The near-infrared transmittance and Dumas combustion (LECO) were applied for the measurement of the nitrogen (N) content to calculate the protein content based on Protein% = N% x 5.7 [102]. The percentage weight of the grain samples that pass through a 2mm sieve/slotter screen was measured to determine grain size.

**2.3.5 Crop water use efficiency.** Water use by crop (WU) was estimated as the difference between the sum of in-crop rainfall and the soil water content at times of sowing and harvest [103], should note that in here soil evaporation is assumed to be part of WU. The water-use efficiency (WUE) was defined as the amount of grain that is produced per unit of water used by the crop, (i.e., WUE = Yield/WU) [103].

## 2.4 Statistical analyses

**2.4.1 ANOVA.** The equality of the variances (homoscedasticity) for the observed variables was assessed using Levene's test. The dataset was then subject to one-way ANOVA with Turkey HSD (honestly significant difference) Post Hoc test to assess the significant impacts of cover crop treatments on soil TOC, POC and POXC, soil water at wheat planting, wheat biomass at grain filling and early maturity, yield, grain protein and screening size. For those attributes that had unequal variances (resulting P-value <0.05 based on Levene's test), Games-Howell test was conducted for nonparametric post hoc analysis. Sources of variation were partitioned into between-group factors (treatment). The mean values of these variables were compared under different cover crop treatments with P<0.05 accepted as being significant.

**2.4.2 PCA.** Kaiser-Meyer-Olkin (KMO) test was used to determine the sampling adequacy of the observed data, with KMO value closer to 1.0 is ideal while values less than 0.5 is considered unacceptable [104]. The KMO value in the acceptable range as it was equal to 0.687. Bartlett's test of sphericity was also applied to test if the observed variables were ideal for factor analysis with P<0.05 being accepted as suitable [105]. Then the dataset was subjected to the principal component analysis (PCA) to interpret our multi-dimension observed dataset and assist with exploring the underlying correlations among observed attributes. IBM SPSS Statistics 27.0 (for Windows) was used for the one-way ANOVA and PCA analysis.

# 3. Results

## 3.1 Soil organic carbon affected by cover cropping

At the trial site, soil total organic carbon (TOC), POC and POXC contents in topsoils (0–10cm) increased at each termination time of the summer cover crop. Early, mid and late terminated plots had greater TOC by 7%, 12%, 17%, and POC by 9%, 24%, 72% in comparison with the control plots (Table 4, Fig 3). POXC contents in early, mid and late termination plots were all lower than the control (Fig 3).

**Table 4. One-way ANOVA tests showing the significant level of the observed variables means under cover crop treatment in comparison to summer fallow (control).** Comparisons without a significant level were considered statistically insignificantly different i.e., P-value>0.05. Grey shadowed values: a significance P<0.05 was observed.

| Observed Items | Dependent Variable | Early Termination | Mid Termination | Late Termination |
|---|---|---|---|---|
| SOC at cover crop termination | TOC | 0.579 | 0.123 | 0.016 |
| | POC | 0.806 | 0.744 | 0.001 |
| | POXC | 0.122 | 0.578 | 0.992 |
| Soil water at wheat sowing | 0-15cm | 0.991 | 0.623 | 0.026 |
| | 15-30cm | 0.611 | 1.000 | 0.0005 |
| | 30-45cm | 0.996 | 0.082 | 0.002 |
| | 45-55cm | 0.994 | 0.705 | 0.0001 |
| | 0-150cm | 0.988 | 0.472 | 0.0002 |
| Wheat biomass during growing season | Biomass, Grain Filling | 0.238 | 0.936 | 0.010 |
| | Biomass, Early Maturity | 0.732 | 0.998 | 0.007 |
| Wheat yield and grain quality at harvest | Yield | 0.728 | 0.755 | 0.001 |
| | Grain Protein | 0.447 | 0.343 | 0.064 |
| | Screening size | 0.041 | 0.902 | 0.917 |
| Soil OC at the end of winter | Fallow TOC | 0.833 | 0.876 | 0.179 |
| | Fallow POC | 0.885 | 0.546 | 0.171 |
| | Wheat TOC | 0.636 | 0.294 | 0.652 |
| | Wheat POC | 0.962 | 0.138 | 0.361 |

At the harvest of the winter wheat which was planted following the summer cover crop (Table 1), greater TOC contents were observed in plots that previously had early, mid and late termination, by 7%, 11%, 7%, respectively, and greater POC by 11%, 52%, 38%, in comparison with plots that were under control during summer (Table 4, Fig 3).

## 3.2 Soil water storage affected by cover cropping

The termination time of the summer cover crop affected the soil water at the sowing time of winter wheat with the greatest soil water observed in early termination plots (Fig 3). The soil water content at 15cm, 25cm and 35cm were lowest at late termination plots compared to the control (Table 4). The soil water of the whole profile (0–150 cm) at wheat sowing was observed to be in the order of the highest to the lowest: control>early termination>mid termination>late termination (Fig 3). While a 1% decrease in whole profile soil water was observed for early termination, soil water increased by 2% in 15cm, 4% in 25cm and 1% in 35cm, compared to the control (Table 4). Mid terminated plots had lower soil water at 15cm, 35cm and across profile by 8%, 10%, and 3%, compared to the control plots, but no difference was observed at 25cm (Table 4). Soil water at 15cm, 25cm, and 35cm and in the whole profile at late terminated plots were lower than the control by 28%, 19%, 18 and 7% (Table 4).

Soil water changed over time in all treatments across the soil profile, but the control plots had the least decline and fluctuation during observations (Fig 4). A decline in soil water was observed for all treatments (Fig 4) suggesting water uptake by the plant, and termination prevented further water loss through transpiration and plant usage. At the end of the summer season and wheat sowing time, early termination had similar or even greater soil water compared to the control (fallow) and significantly greater than other treatments (Fig 4). As shown in Fig 5 there was no significant rainfall two months before wheat planting, but early termination was able to store the received rainfall. In comparison with control plots, soil water contents in mid and late terminations plots were both affected by the delayed

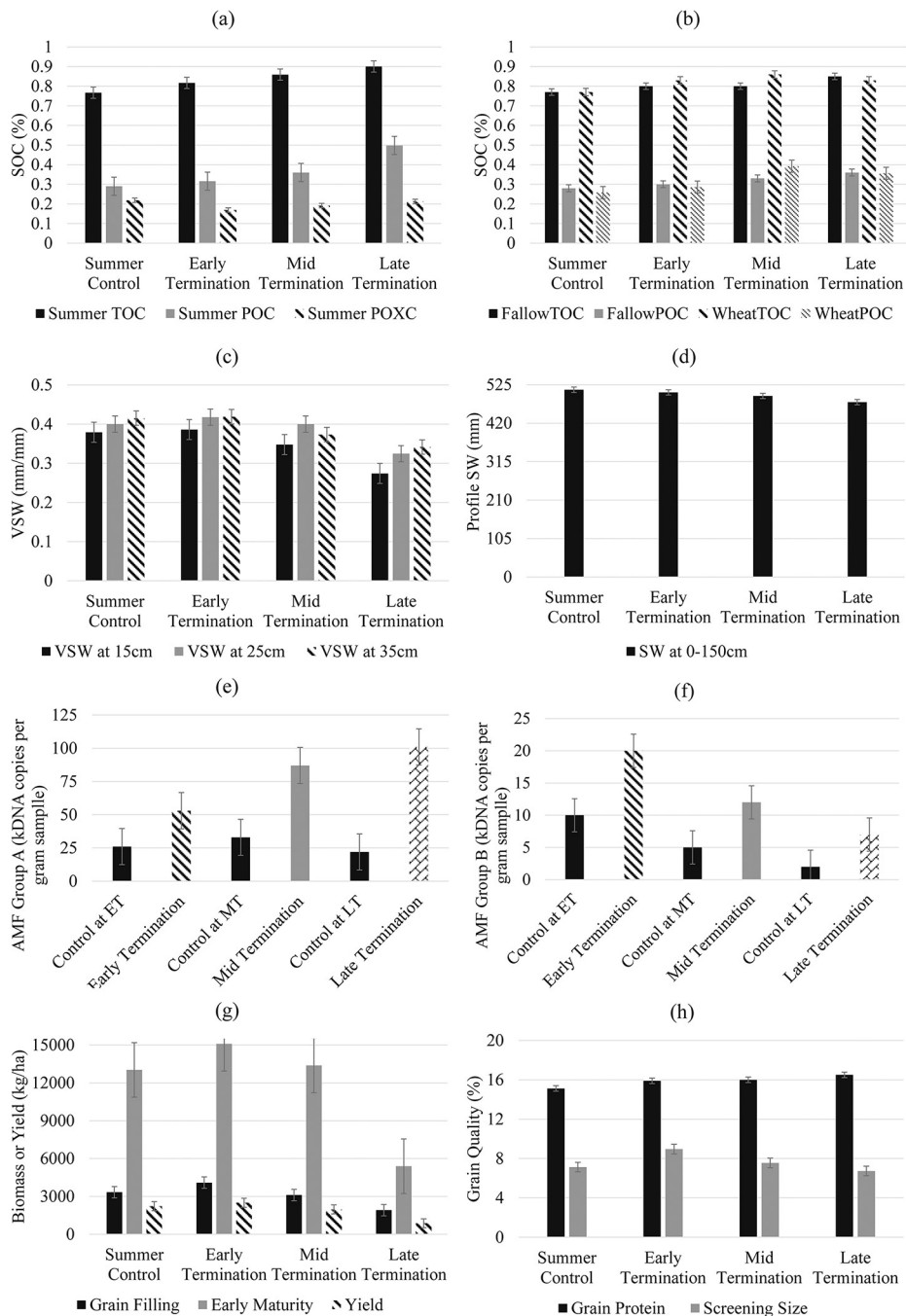

**Fig 3.** (a) Soil total organic carbon (TOC) and particulate organic carbon (POC) and permanganate oxidizable carbon (POXC) at 10 cm depth at the time of termination of summer cover crop; (b) TOC and POC measured at harvest time of winter wheat; (c) volumetric soil water (VSW) content at 15cm, 25cm and 35cm and (d) whole profile soil water at wheat sowing time; AMF Group A (e) and Group B (f) measured at the time of termination of summer cover crop; (g) wheat above-ground dry biomass measured at grain filling and early maturity phenology stages and yield at harvest; (h) grain quality. SC: Summer control; ET: Early termination; MT: Mid termination; LT: Late termination.

termination, especially in deeper soil layers and the whole profile (Fig 4). There was a 14% decline in the whole profile under late termination compared to the control. Overall, mid and lateterminated cover crop did not show an advantage in preserving greater soil water

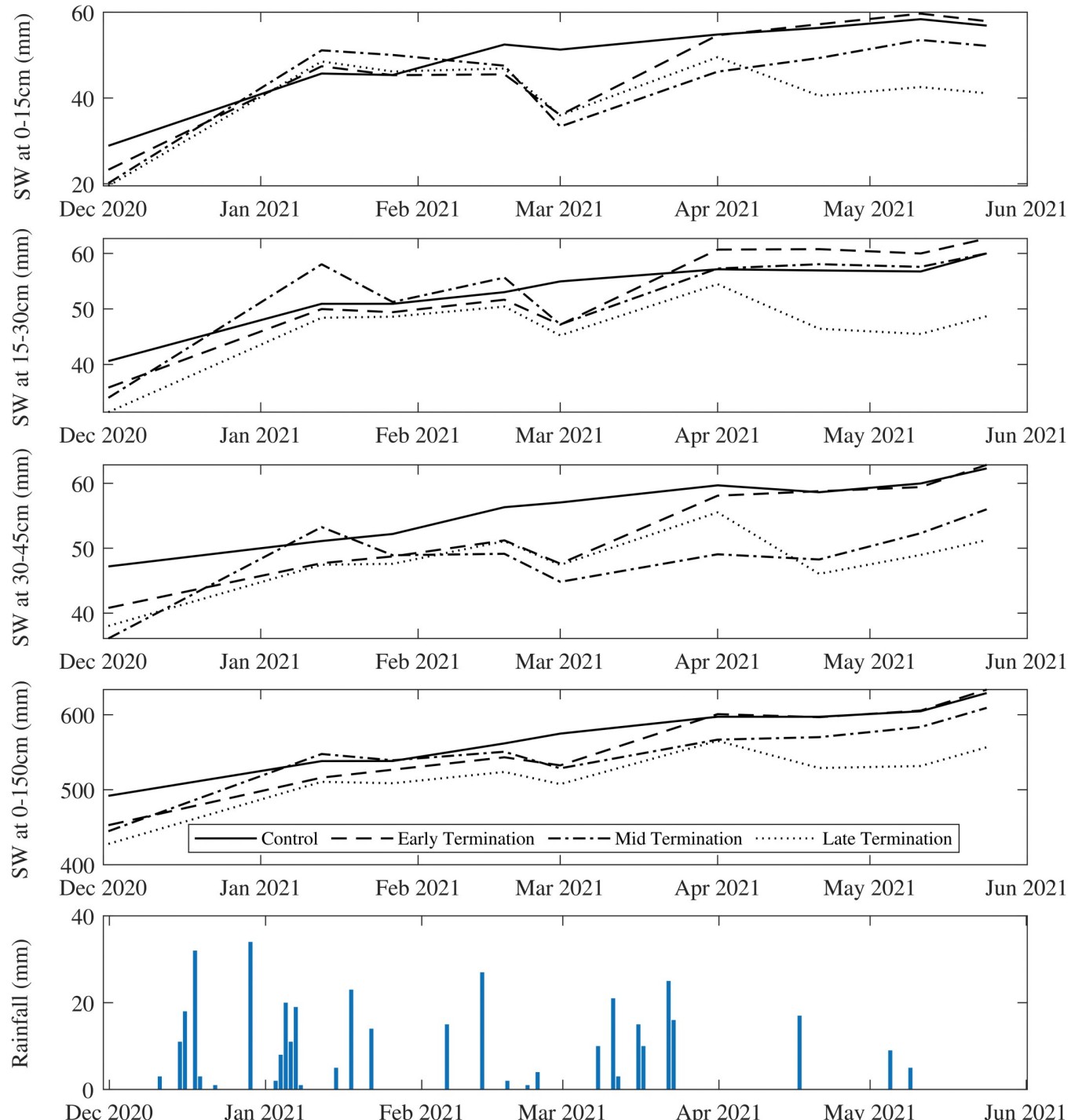

**Fig 4. Soil water storage of top layers (0-15cm, 15-30cm and 30-45cm) and whole profile (0–150 cm) during summer cover crop.** At the planting time, in the absence of rainfall for 18 days before planting, Early termination had stored higher water in surface soil (0-30cm) while performed same as fallow for all other layers and averaged for the whole profile.

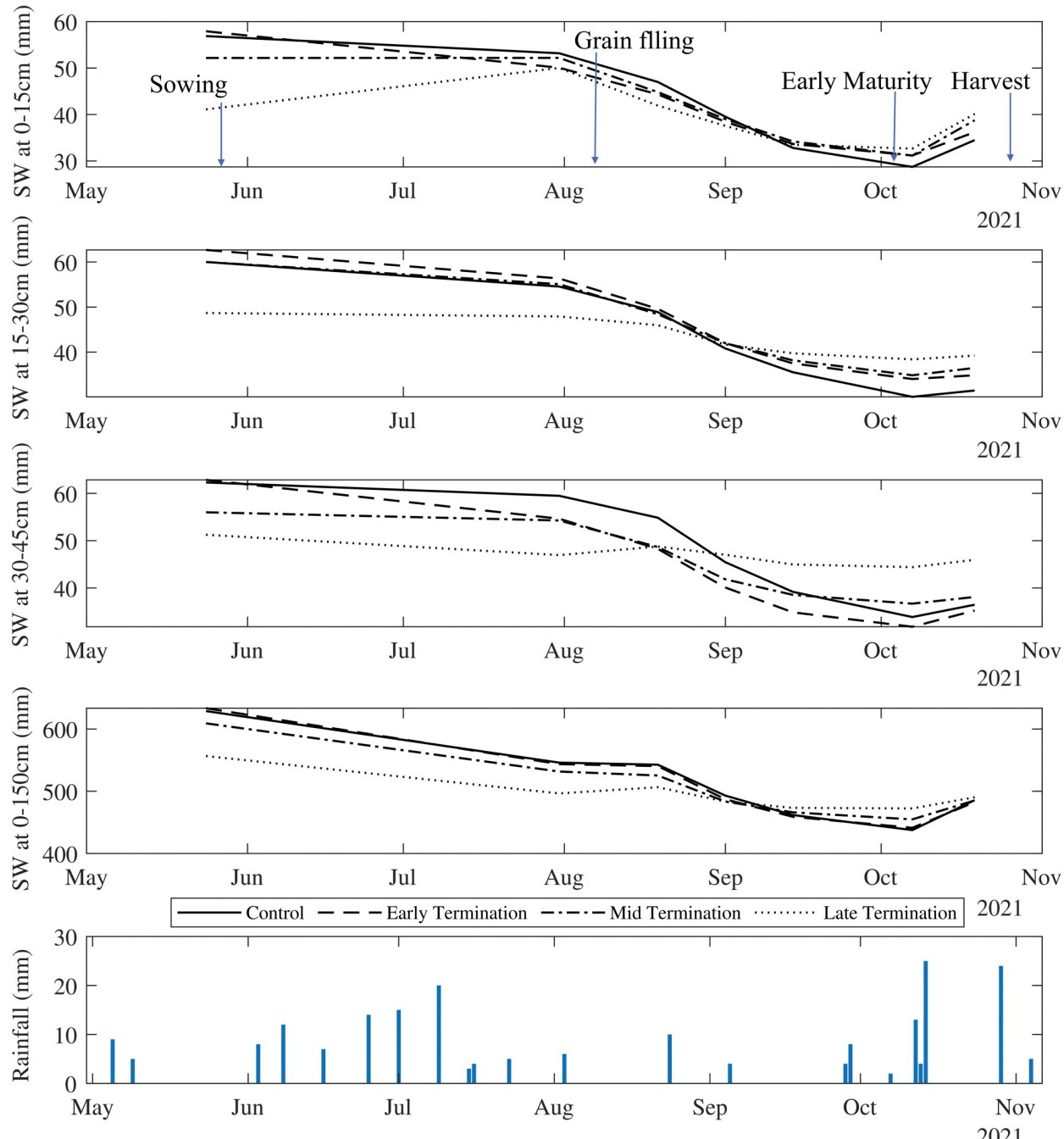

**Fig 5. Soil water in topsoil layers (0–15cm, 15–30cm and 30–45cm) and whole profile (0–150cm) during the winter season.**

than the control at wheat sowing time, possibly due to little rain received after termination and prior to planting.

**3.2.1 Changes in soil water storage over winter cover crop.** Soil water in the surface layer (0–15 cm), where wheat was planted on early terminated summer cover crop plots, did not differ significantly over the season when compared to those planted in control plots (Fig 5). Fallow, early, and mid-terminated plots had almost similar soil water at the grain filling

stage across the whole soil profile, but soil water was higher in 15–30cm depth under early termination (Fig 5). However, soil water was significantly lower at sowing time in plots where summer cover crop was treated with mid and late termination.

At the flowering stage, wheat plots that were planted on early terminated summer cover crop plots had 0.5% less profile soil water compared to the control, followed by 2% less in mid-termination and 9% less in late termination. When the wheat crop reached initial grain-filling stage, the difference in profile soil water compared to the control were -0.4%, 1% and -7% in wheat plots following early, mid and late terminated cover crops. At the end of grain filling, the whole profile's water differences were lower than the control by 1.8%, 2.1% and 2.5% in wheat plots following early, mid, and late termination. Wheat planted on late termination had the lowest soil water content until the grain filling stage compared to the control and the other treatment plots, however, for 0–15cm layer, soil water was not significantly different to other treatments at the flowering stage. By the end of winter (harvest rips), the highest soil water contents for top layers and whole profile were observed in wheat plots following late termination, followed by mid and then early termination.

**3.2.2 Cover crop and water use and -use efficiency.** Cover crops were terminated at different dates, so the amount of WU by crop and in-season rainfall received at each termination treatment was different (Table 5). The early terminated cover crop plots had the opportunity to receive the least in-season rainfall during growth (91mm), therefore this treatment had less opportunity to use water (73.1mm) compared to the other treatments (Table 5). Contrary, late termination plots used 187.8mm from 208mm of rainfall that they received. In winter, all wheat plots received the same amount of rainfall (~164mm), but their WU and WUE varied in different plots due to the effect of the previous cover crop treatment in summer (Table 5). Wheat planted on early termination cover crops had the highest WU and WUE compared to the wheat planted on summer control by 2% and 10%. The wheat planted on mid termination plots had 2% greater WUE, though its WU was 14% lower than the wheat planted on summer control plots. The wheat planted on late termination plots had lower WU and WUE than the wheat planted on summer control plots.

## 3.3 Arbuscular mycorrhizal fungi affected by cover cropping

In control plots, AMF group A DNA sequence concentration (SC) increased from the time of early termination towards the time of mid termination but then declined at late termination (Fig 3). The greatest AMF Group A SC was observed in late termination plots, followed by mid

**Table 5. Water use (WU) by summer cover crop and water use efficiency (WUE) in wheat plots following cover crops.**

| Variables | Summer Cover Crop Treatment | | | Late Termination |
|---|---|---|---|---|
| | Control (Fallow in summer) | Early Termination | Mid Termination | |
| *Summer cover crop season* | | | | |
| Rainfall, planting to termination (mm) | - | 91 | 191 | 208 |
| WU (mm) | 0 | 73.1±16.9 | 170.3±7.9 | 187.8±13.2 |
| Rainfall, termination to planting (mm) | - | 131 | 31 | 14 |
| *Wheat planted following summer cover crop* | | | | |
| Rainfall, planting to harvest (mm) | 164 | 164 | 164 | 164 |
| Wheat yield (kg/ha) | 2226±219 | 2500±221 | 1965±154 | 870±133 |
| WU (mm) | 289.7±13.2 | 296.8±10.1 | 250.5±14.3 | 178.3±10.1 |
| Wheat WUE (kg/ha.mm) | 7.7±0.7 | 8.5±1.0 | 7.9±0.8 | 4.8±0.6 |
| ΔWU, Treatment Vs planted on summer control | - | 2% | -14% | -38% |
| ΔWUE, Treatment Vs planted on summer control | - | 10% | 2% | -37% |

and early termination (Fig 3). AMF group B SC in control and cover cropped plots all decreased from the time of early termination towards the time of late termination. Overall, DNA SC of both AMF Group A and B were different between the treatment plots and control plots. AMF Group A DNA SC increased by 356% in late termination plots, 162% in mid termination plots and 104% in early termination plots compared to the control. DNA SC of AMF Group B in late termination plots was 251% greater than the control, 119% greater in mid termination and 100% greater in early termination.

## 3.4 Wheat biomass, yield and grain quality

During the winter season, wheat above-ground biomass was monitored at two different growth stages, with the first biomass samples collected during the grain filling stage and the second biomass samples collected during early maturity. Observations showed that wheat above-ground dry matter from late termination plots was 43% and 59% lower than the control plots, in two biomass sample observations. The biomass in early terminated plots was 23% greater at the grain filling stage and 16% greater at early maturity compared with control plots. The biomass of mid termination plots was 7% lower at the grain filling stage but 3% greater during early maturity compared to the control plots.

The grain yield was highest in early termination plots i.e., 12% higher than control. The yield under mid and late termination treatments was 12% and 61% lower compared to the control (Table 6, Fig 3). Among all wheat plots, the highest grain protein content was observed in late termination plots (Fig 3). Grain protein from late termination plots was 9% higher than control, while early and mid-termination plots had 5% and 6% higher grain protein (Table 6). Grain screening size in early termination plots was 26% higher than in control, followed by mid termination (6%), but late termination plots had 6% lower screening size (Table 6).

The one-way ANOVA results showed a significant difference between the variance of cover crop treatment and the control (Table 4). Soil TOC and POC contents measured at termination times of summer cover crop showed a significant difference between late termination and the control (P-value = 0.016 and 0.001). No significant difference in soil POXC content was

**Table 6. Relative change in soil carbon, soil water, wheat biomass, yield and grain quality affected by summer cover crop compared to the control (fallow).**

| | | Summer cover crop treatment | | |
|---|---|---|---|---|
| Plots | Variables | Early Termination | Mid Termination | Late Termination |
| Summer cover crop | Δ TOC | 7% | 12% | 17% |
| | Δ POC | 9% | 24% | 72% |
| Winter control | Δ TOC | 4% | 4% | 11% |
| | Δ POC | 10% | 19% | 30% |
| Winter wheat | Δ TOC | 7% | 11% | 7% |
| | Δ POC | 11% | 52% | 38% |
| Crop after summer cover crop | | | | |
| Wheat, sowing time | ΔVSW, 15cm | 2% | -8% | -28% |
| | ΔVSW, 25cm | 4% | 0% | -19% |
| | ΔVSW, 35cm | 1% | -10% | -18% |
| | Δ Profile SW | -1% | -3% | -7% |
| Wheat, during season and harvest time | Δ Biomass, grain filling | 23% | -7% | -43% |
| | Δ Biomass, early maturity | 16% | 3% | -59% |
| | Δ Yield | 12% | -12% | -61% |
| | Δ Grain Protein | 5% | 6% | 9% |
| | Δ Screening Size | 26% | 6% | -6% |

found between treatments and the control in summer cover crop. Wheat above-ground dry biomass collected during grain filling and early maturity phenology stages showed significant differences between plots with a history of summer late termination and the control (P-value = 0.01 and 0.007). Wheat yield in plots with a history of late termination was also significantly lower compared to the grain yield from the summer control plots (P-value = 0.001).

No significant difference was observed in grain protein content between plots with cover crop treatments and the control. Plots with a history of early termination had significantly higher grain screening size (P-value = 0.041) in comparison with control plots (Table 4). Late termination exhibited significantly less soil water across the profile (0–150cm) at wheat planting compared to the control (P-value = 0.026). Overall, the late-terminated cover crop disadvantaged preserving soil water, and consequently affected grain yield, though it was able to significantly increase TOC and POC by termination time.

### 3.5 Relationships among the variables

PCA results (Fig 6) showed that within the dimension of component 1 (34.6% of variance), wheat yield and biomass were closely related to soil water at 15–30cm and 0–150cm, especially in plots with a history of the early terminated cover crop during summer, followed by soil water in 30–45cm and 0–15cm. In component 2 (19.4% of variance), PCA revealed an underlying correlation between soil OC contents (TOC, POC and POXC) and clay content (Fig 6). PCA did not exhibit an underlying relationship between grain quality and the other observed variables. Overall, an underlying correlation of OC with soil clay content and yield with soil water at planting time was observed.

### 3.6 TOC and Labile OC relationships

Results show that soil POC had a relationship with TOC content, and a greater correlation between soil TOC and POC was found in summer cover crop plots and wheat plots under

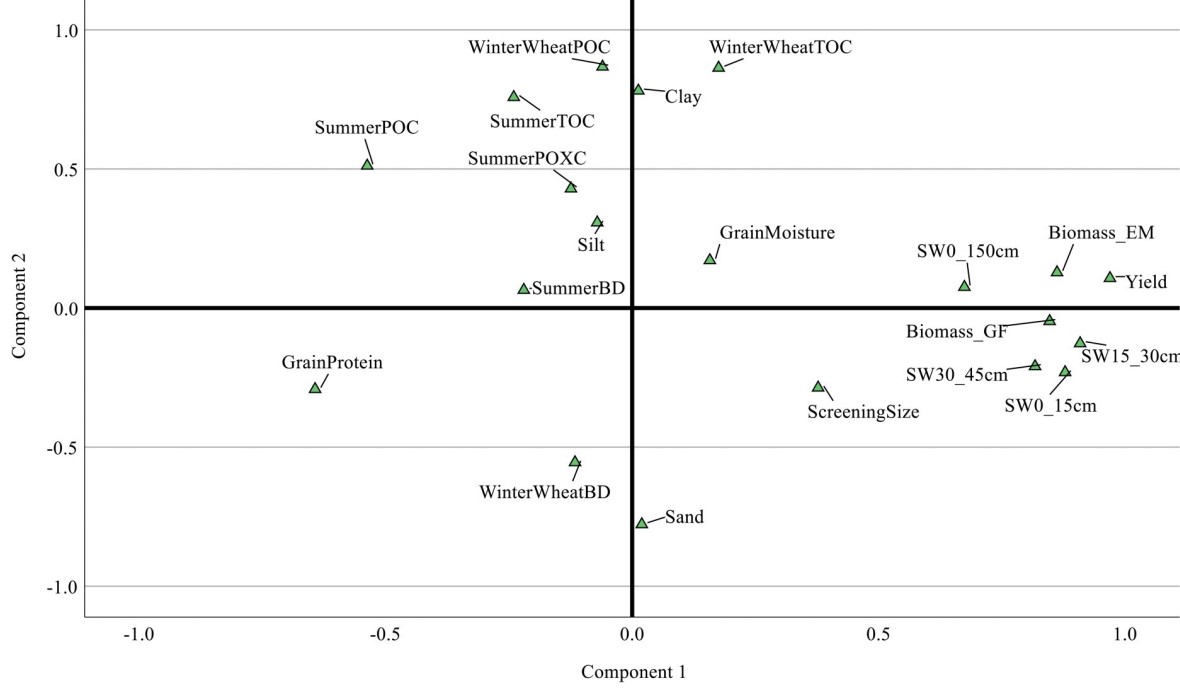

**Fig 6. The two-dimensional principal subspace for the observed data.** SW: Soil water; BD: Bulk density.

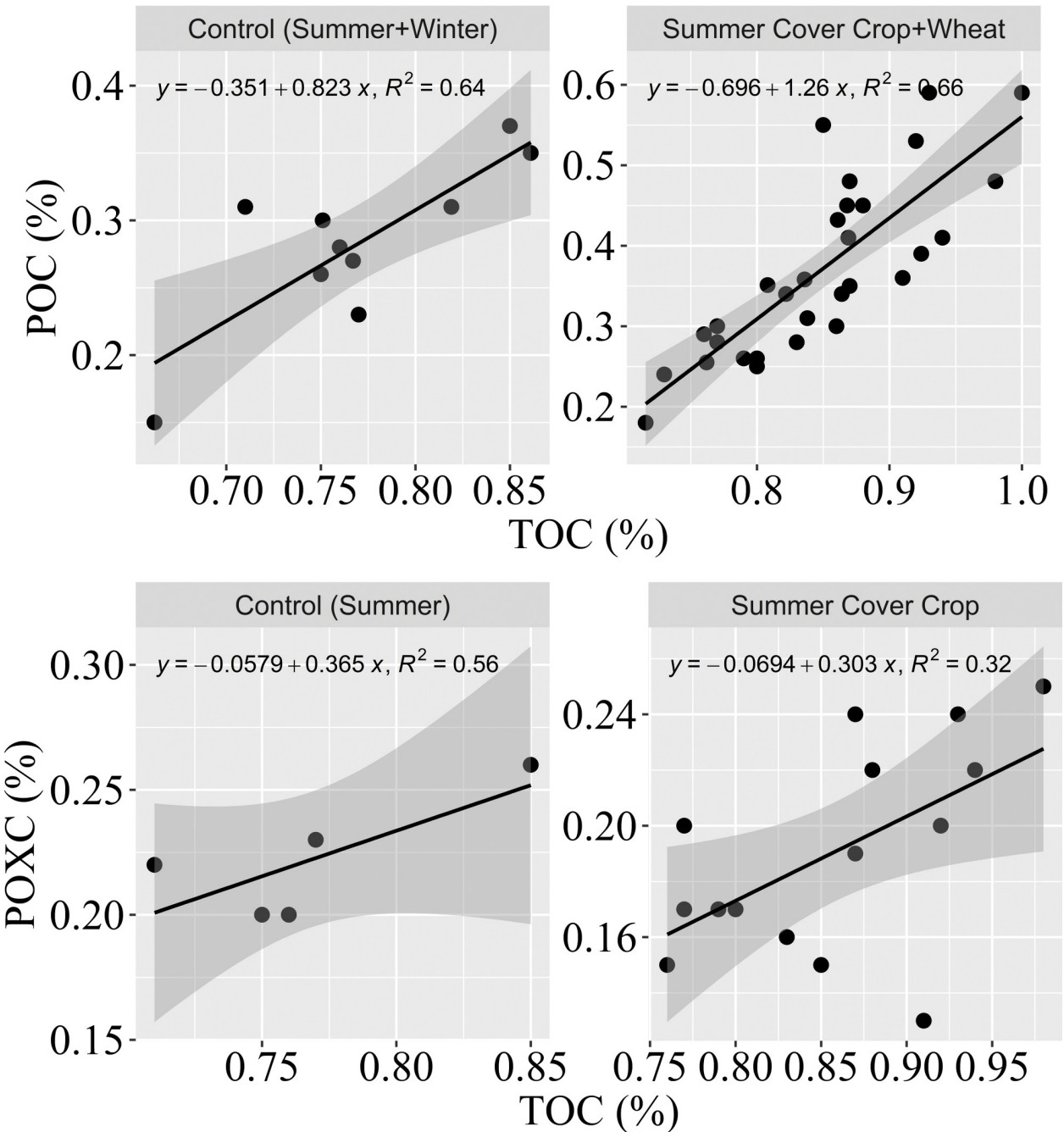

**Fig 7. The relationship between total organic carbon and particulate organic carbon (POC) and permanganate oxidizable carbon (POXC).**
Top left: TOC and POC in summer and winter control plots; Bottom Left: TOC and POXC measured in summer control plots; Top right: TOC and POC in summer and winter cover crop plots; Bottom right: POXC and TOC in summer cover crop plots. The grey shadow area represents a 95% confidence interval.

previous cover crop treatments compared to the control (Fig 7). Meanwhile, results showed that POXC and TOC in summer had a relationship in control plots. However, the relationship between POXC and TOC was not strongly correlated in cover crop plots (Fig 7). Overall, the relationships between TOC and two labile OC fractions were different in cover crop plots and control plots, with TOC accumulation being more sensitive to the increase of POC, especially

under the impact of cover crop management (i.e., the presence of cover crop and how long till termination).

## 3.7 Yield affected by soil water at wheat planting

Soil water at the sowing time of cash crops is critical to seed establishment and biomass production. Soil water at surface layers 0–15cm, 15–30cm and 30–45cm and soil profile water (0–150cm) affected the wheat above-ground dry biomass at both grain filling and early maturity stages (Fig 8). Results also showed that the surface layer 0–15cm stored soil water at wheat sowing had a greater effect on yield, compared to the soil water in the whole profile (Fig 8). Wheat planted on early termination plots had the highest yield, while for those planted on late termination, the yield was the lowest. Fig 8 shows that summer cover cropping practice through managing the termination dates impacted soil water availability at the planting of winter crop, which affected the crop's above-ground biomass accumulation and yield.

## 4. Discussion

Incorporation of cover cropping into a crop-fallow system has been practised as a means to manage ground cover, organic matter, stored soil water, soil quality and health. In this research, the legacy impact of summer cover crops on soil water across soil profile was explored and our results demonstrated the effectiveness of replacing summer cover crop with fallow. ANOVA test showed the significant effect of treatments on stored water, TOC, POC, wheat biomass, yield and grain size.

For the examined season, the early termination of summer cover crop resulted in a 2%, 4% and 1% increase in soil water at planting time at depths of 0–15cm, 15–30cm and 30–45cm, respectively, and subsequently led to a 12% increase in wheat compared to the control. Additionally, this treatment increased TOC and POC levels by 7% and 9%, respectively (Table 4). The summer cover crop was found to enhance soil biology as evidenced by an increase in AMF sequence concentrations in both A and B groups (Fig 3C and 3F). Managing summer cover crop effectively could potentially increase soil water storage during the growing season of the winter cash crop, which could be crucial for sensitive phenology stages (Fig 5). While an underlying correlation was observed between soil water, biomass and yield (Fig 6), the combined effect of changes in soil water and organic carbon resulted in an increased yield (12%) and improved quality, such as a 5% increase in grain protein (5%) in the early-terminated cover crop treatment (Table 4).

### 4.1 Cover cropping affects soil-water relations

In the current study season, soil water content was affected by including summer cover crops and by the timing of termination with greater differences observed in the top layers (0–15cm, 15–30cm and 30–45cm) than in the whole soil profile (0–150cm) (Fig 4). Regardless of the soil water loss due to plant water use, the evidence from trials suggests that the inclusion of cover crops with optimal termination and residue retention creates a beneficial legacy within the soil profile. The early termination was able to retain more soil water compared to the other scenarios (Fig 5) while enhancing chemical and biological indicators i.e., TOC, POC, AMF sequence concentration (Fig 3). This finding provides evidence that cover crop with optimal termination could maximise the soil water storage for a dryland cropping system in the Northern Grain Belt region of Australia. These results apply to our study site for a season with relatively low rainfall but the effect of cover crop management on soil water storage could be different depending on climate variability, soil condition and management [35]. However, this study provides sufficient evidence that cover crop management and its impact on soil health played a

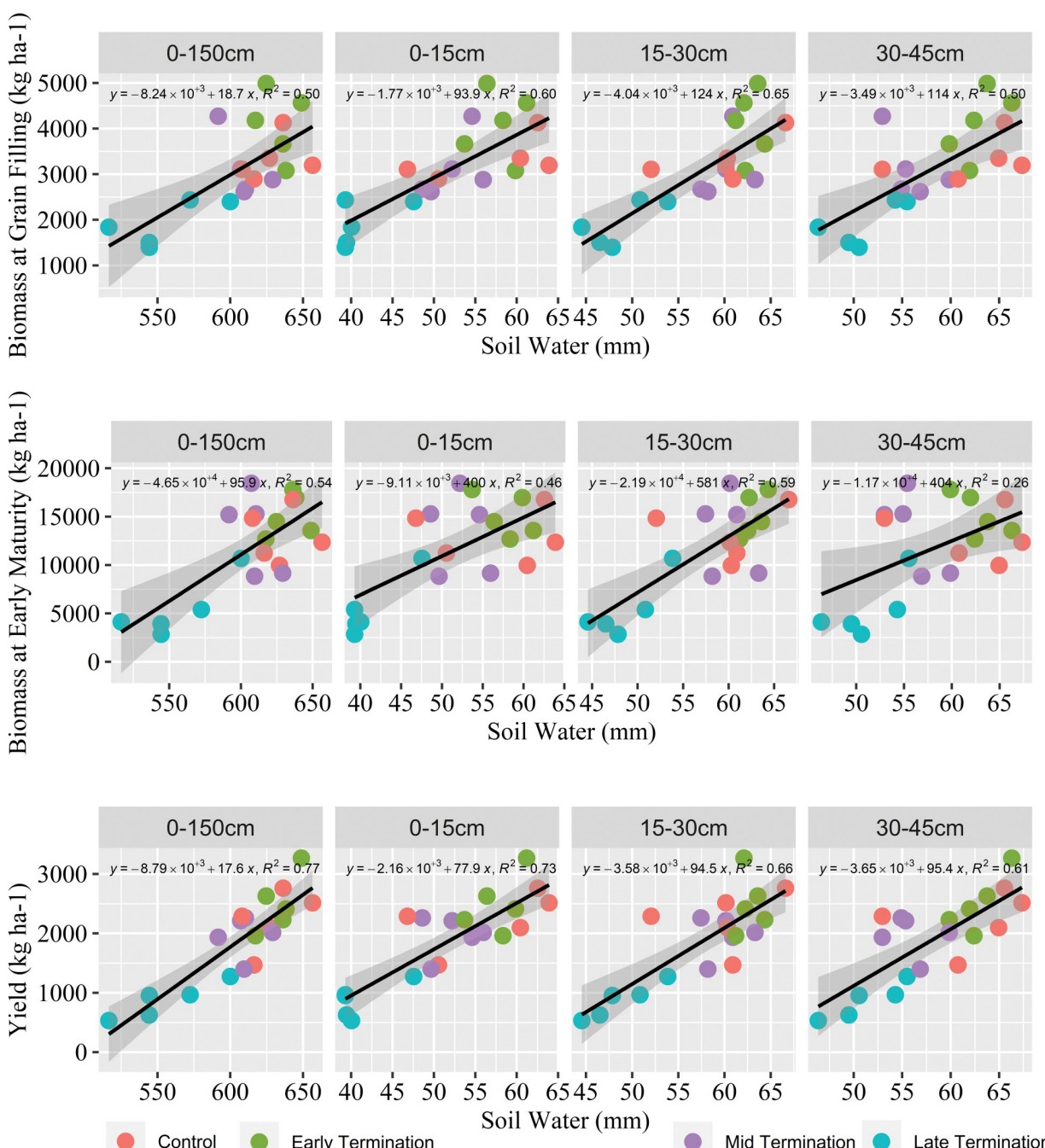

**Fig 8. Relationship between soil water at top layers and whole profile at planting time and wheat yield and biomass accumulation at grain filling and early maturity stages.** The grey shadow area represents a 95% confidence interval.

substantial role in soil water storage and yield. Early termination had similar soil water content at wheat sowing with the control, but enhancements in soil health indicators (i.e., OC and microbial activity) played a role in the increase in yield and grain quality that was observed. Overall, the evidence from trials suggests that the inclusion of cover crop with optimal termination and residue retention can be effective in retaining soil water while contributing to

improved soil biological health (e.g., increased organic matter and microbial activities) and yield production of the following winter crop.

## 4.2 Available soil water at planting as a driver of wheat biomass and yield

Winter wheat biomass and yield associated with summer cover crop were attributed to available soil water at sowing time, evidenced by its correlation with wheat above-ground biomass at the grain filling stage and early maturity stage (Fig 8). Wheat biomass during grain filling and early maturity response to stored soil water at sowing time was negatively affected by the mid and late termination of cover crop (Fig 8), due to inadequate soil water at sowing contributed to a reduction/delay in crop establishment and biomass production. Despite the increase in soil OC and microbial activity, the cause of such reduction in wheat biomass and yield was associated with the incorporation of longer summer cover crop treatments i.e., mid and late termination. The summer cover crop was terminated and then left standing, which allowed the soil surface to have ground cover even after the termination. However, mid and late- terminated plots had greater above-ground biomass and hence crop residue after termination. A physical barrier of the heavy/dense cover crop residue may lead to an unfavourable/adverse impact on wheat emergence by obstructing light penetration and releasing phytotoxic chemicals from the residue. This phenomenon of crop residue inhibiting plant emergence was also reported in other field studies [106, 107].

As shown in Fig 5, summer cover crop treatments (i.e., early termination) could provide soil water similar to summer control at the grain filling stage while facilitating soil biological activities. Soil water availability is critical to wheat root growth and above-ground biomass accumulation, especially for rainfed crops during the grain filling stage as wheat has a higher water uptake rate at this stage [108]. Biomass at maturity can affect the final grain production of wheat [109]. Furthermore, dryland wheat growth, grain yield and quality are highly dependent on the amount of soil water storage at the planting, flowering, and grain-filling stages [110–112]. Field data represents a typical Australian dryland cropping system where the availability of soil water storage and water use efficiency are limited [113]. The previous research studies stated that lower soil water availability at planting can lead to a decrease in wheat yield as affected by the incorporation of cover crops. [114, 115], here the finding of this study further highlights the importance of cover crop management and shortening of summer fallow (also called short fallow).

Greater wheat biomass production following early termination of cover crop contributed to higher water use efficiency of winter crop, and consequently greater yield production (Table 6). This was due to the combined effect of increased soil water at sowing, soil OC, and microbial activities, as discussed in 4.1. With increasing concern about climate change and droughts, the availability of water resources is becoming crucial to dryland cropping systems and system WUE which is often used as a target for soil management [115]. This study indicates that managing soils through proper cover crop management can improve WUE and potentially crop biomass and yield. Cover crop management can be practised for improving productivity via enhanced soil water storage and WUE which can be helpful in facing the challenges of climate change and drought events.

## 4.3 Cover crop affecting soil organic carbon

Soil responses to summer cover crop, specifically soil TOC, POC and POXC were different under cover crop treatments and fallow. In summer, the greatest soil TOC at cover crop termination was observed in the soil surface layer (0–10cm) of the late termination plots and was significantly greater than the TOC in control by 17% (Table 4), which could be an outcome of

developed root systems in soils and enhanced biological activities observed by an increase in AMF DNA SC (Table 4). Soil POC content at cover crop termination differed among treatments, with the most significant difference (also the greatest) observed in late termination plots, it was likewise associated with increased soil AMF activities (Table 4). Soil POC constitutes hotspots for microbial activities and has been used as an indicator of soil biological activity [116]. With enhanced soil AMF growth and activities in the late termination plots, AMF was able to facilitate fresh residue decomposition and increase POC availability [54]. Soil POC was considered a performance indicator for changes in soil quality [117–119] showed that variation in POC can account for 69–94% of the changes in TOC due to different land use and management. Various other studies have reported similar findings regarding cover crops of different species improving soil TOC and POC contents [118, 120]. Across all treatments, TOC and POC were correlated to each other (Fig 7) suggesting: 1) cover crop management had a consistent effect on improving both TOC and POC availability compared to the control; 2) an increase in POC content contributed to increase in TOC pool.

Different from soil TOC and POC, results showed that POXC at cover crop termination was the greatest in the control plots, followed by late, mid and early plots, but the differences among treatments and control were not significant. This suggests that cover crop management did not significantly affect POXC content over the short term and control plots had a simpler system where POXC was probably not decomposed/utilised by soil microorganisms as faster as the soils in cover crop plots [121–123]. Some studies reported that POXC was sensitive to management practices and could be used as an early indicator of improved soil organic matter management [44, 124, 125] but cover crop treatments sometimes can have little effect on POXC due to low content of soil organic matter [126]. Both POC and POXC are the measurements of labile organic carbon, the POC method was found to be more sensitive to rapid gain in OC as a result of management or land-use change, while POXC was found to be more sensitive to soil lignin content (a stable component of SOM), instead of rapid gains in OC [127]. Based on our results, soil POC was more correlated with changes in TOC while less correlation was found between POXC and TOC (Fig 7). This may suggest POC in our experiments was sensitive to the changes in TOC due to cover crop incorporation. While, as POXC was sensitive to changes in soil lignin compounds which were sourced from surface residue decomposition. Our finding also suggests that: 1) POXC was particularly insensitive to the changes in TOC likely because the trial site had crop residue retained from the previous years and the crop residue has not been fully decomposed at the time of early or mid termination and hence there was little lignin input in these two treatment plots; 2) late termination treatment allowed more time for the residue to decompose (including the wheat stubble from the previous year and fallen litter from the cover crop), and consequently had more lignin input and stimulated POXC accumulation.

Soil OC components measured at the end of the winter season showed that wheat planted on mid termination plots had an advantage in storing more TOC and POC by 11% and 52% compared to the control but disadvantaged yield by 12% compared to the control (Table 4). This was likely a result of a better soil water-microbial environment to handle residue retained and input into the soil compared to the other plots. Overall, results of this study showed that the short-term cover cropping in summer promoted a rapid gain in soil TOC and POC.

Based on existing studies, the positive relationship between soil OC and crop yield begins to level off when soil OC content reaches approximately 2% [128, 129]. However, no potential correlation between soil OC and yield was observed in trial's soils as OC content was below 1%, which may not be the sole factor driving the grain yield.

### 4.4 Cover cropping affects arbuscular mycorrhizal fungi groups differently

The results showed different DNA sequence concentration of soil AMF Group A and B at termination time, and their response to fallow and cover crop treatments varied (Fig 7). This was consistent with the study of [130] that reported AMF species had different root colonization rates depending on the AMF family (taxonomic variation). The response to AMF colonisation differs in host plant species, root growth and the space available for root development [131]. Each group's species may have a similar response to the changes in environmental factors such as variation in soil properties and host plant biomass [132, 133] which can occur under cover cropping. The presence or absence of AMF colonisation is also related to soil water conditions which in our trials in vertosols, soil fluctuates seasonally to favour or hinder the AMF associations with the host plants [134].With greater AMF Group A DNA sequence concentration found in late termination soil compared to the control, it was likely because late termination plots had greater sorghum root biomass, which allowed a higher chance for AMF Group A to colonize and establish [62]. The greatest AMF Group B DNA sequence concentration was found in early termination plots, compared to the control, and the lowest was found in late termination plots. The decreasing pattern of AMF Group B DNA sequence concentration from the time of early termination towards late termination was possibly related to soil water availability in the rhizosphere zone [135, 136]. In addition, previous works suggested that intense competition among AMF over root space could lead to competitive dominance in the colonization of some AMF species by excluding others [137, 138].

### 4.5 Limitations and recommendations

Overall, this study was subject to potential limitations. Findings of this study were based on the trials within a 1-year window, although with sufficient replications and two examined seasons that had a relatively typical rainfall (Fig 1), a longer term observation might be needed. Our on-farm cropping system research aimed to explore the plant-soil-water relations with implications of summer cover crop practice over the growing seasons of cover crop and cash crop. This has certain significant values to future field studies in the eastern region of the wheat belt as soil water at planting plays a critical role in cash crop establishment and yield production. The changes of POC (which is responsive to short-term management change) did capture the impact of summer cover crop and suggested an improvement of soil quality related to SOM and microbial activities. Therefore, it is recommended that summer cover crop incorporation could help to promote soil health (organic carbon accumulation and microbial activities) through residue retention. This study also showed the importance of the timing of cover crop termination, for its impact on soil water storage. For future studies, it is crucial to consider a number of factors prior to implementing cover crop practices: 1) decision on planting and termination of cover crops should be carefully planned; 2) considering the impacts of cover cropping because it may not necessarily achieve all the benefits (adequate soil water preservation, yield increase, carbon accumulation, microbial health enhancement) that can be affected by many factors such as soil condition, growing season, climate, management and investment decisions; 3) considering the potential impact of cover cropping on soil nitrogen retention and their regulation effects on nitrogen cycling processes. For assessing long-term effect of cover cropping practice, it is also recommended to apply validated biophysical modelling to investigate the interactions between soil-crop under the effect of climate variability and management.

## 5. Conclusion

The implementation of summer cover crop with early termination improved soil biological health and increased soil water content at wheat sowing time, which collectively enhanced

wheat WUE, yield and grain protein content. This study also highlights the importance of timely termination and residue retention. Cover crop with late termination had some drawbacks such as depleting soil water during the growing season and consequently affected soil water availability at wheat sowing time. Although there was evident advantage in soil OC addition and AMF growth under late termination treatment, the loss of soil water at sowing time was detrimental, which led to a significant decline in wheat biomass and yield production. There was a 4% increase in surface soil water at winter wheat sowing time under optimum summer cover crop, but the effectives were not proportional in yield increase i.e., 12% which suggests that yield increase could be benefited from enhancement in soil health i.e., soil OC and potentially microbial activities. Overall, summer cover crop practice showed great potential to increase soil health and crop productivity in dryland agricultural systems. Cover crop can be used to manage soil water and soil health, although further research is needed to consider the climate variability and management regime that will maximise the potential and effectiveness of cover crop practice.

## Acknowledgments

We would like to express our thanks of gratitude to David Lawrence, David Freebairn, Lukas Van Zwieten, and Terry Rose for their contributions to the discussions. The authors would also like to thank Makhdum Ashrafi, Renier Snyman, James Henderson and Luke Laherty of the Department of Agriculture and Fisheries for their support in conducting field works.

## Author Contributions

**Conceptualization:** Hanlu Zhang, Afshin Ghahramani.

**Data curation:** Hanlu Zhang, Afshin Ghahramani, Aram Ali, Andrew Erbacher.

**Formal analysis:** Hanlu Zhang.

**Funding acquisition:** Afshin Ghahramani.

**Investigation:** Hanlu Zhang.

**Methodology:** Hanlu Zhang, Afshin Ghahramani.

**Project administration:** Afshin Ghahramani.

**Supervision:** Afshin Ghahramani.

**Writing – original draft:** Hanlu Zhang, Afshin Ghahramani.

**Writing – review & editing:** Hanlu Zhang, Afshin Ghahramani, Aram Ali, Andrew Erbacher.

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
