## [Decision Letter · Decision Letter 0]

15 Nov 2022

PONE-D-22-23600Cover cropping impacts on soil water and carbon in dryland cropping systemPLOS ONE

Dear Dr. Zhang,

Thank you for submitting your manuscript to PLOS ONE. After careful consideration, we feel that it has merit but does not fully meet PLOS ONE’s publication criteria as it currently stands. Therefore, we invite you to submit a revised version of the manuscript that addresses the points raised during the review process.

We look forward to receiving your revised manuscript.

Kind regards,

Muhammad Riaz

Academic Editor

PLOS ONE

Journal Requirements:

    a. You may seek permission from the original copyright holder of 1 to publish the content specifically under the CC BY 4.0 license. 

“This work has been supported by the Cooperative Research Centre for High Performance Soils whose activities are funded by the Australian Government’s Cooperative Research Centre Program, along with support from both the Queensland Department of Agriculture and Fisheries and the University of Southern Queensland through the Broadacre Cropping Initiative.”

“AG received funding support provided by The Cooperative Research Centre for High Performance Soils (Project No. 2019/4_s.002

https://soilcrc.com.au/) and The Broadacre Cropping Initiative (https://www.unisq.edu.au/research/institutes-centres/ilse/baci).

Additional Editor Comments:

Three reviewers have provided feedback on your manuscript. They have suggested that your manuscript requires revision before further consideration for publication. I invite you to revise your manuscript by giving due considerations to comments and suggestions from reviewers.

Reviewers' comments:

Reviewer's Responses to Questions

**Comments to the Author**

1. Is the manuscript technically sound, and do the data support the conclusions?

Reviewer #1: Yes

Reviewer #2: Yes

Reviewer #3: Yes

2. Has the statistical analysis been performed appropriately and rigorously? 

Reviewer #1: Yes

Reviewer #2: No

Reviewer #3: Yes

3. Have the authors made all data underlying the findings in their manuscript fully available?

Reviewer #1: No

Reviewer #2: Yes

Reviewer #3: Yes

4. Is the manuscript presented in an intelligible fashion and written in standard English?

Reviewer #1: Yes

Reviewer #2: No

Reviewer #3: Yes

5. Review Comments to the Author

Reviewer #1: Cover cropping impacts on soil water and carbon in dryland cropping system is an interetsing study but if authors can suggest how much cover crop will help to incraese soil water and C in dryland cropping system it would be great as numbers ae missing.

Is terminations of summer cover crop practices is common practice ?

Recommenadtions seems not clear as its general please be specific.

Figure quality is also poor.

Reviewer #2: Zhang et al. investigated the effects of replacing conventional fallow period with summer sorghum cover crop and its termination timing in SW QLD on soil water and carbon dynamics, AMF abundance and subsequent winter wheat. The manuscript indicates early termination of summer cover crop improve AMF, soil water accumulation at wheat planting, and wheat grain yield, % protein and WUE. It is quite interesting to figure out how cover cropping can affect soil C cycling, soil water balance and subsequent crop yield in dryland environments. The topic of this manuscript is of interest of the readership of PlosOne, as well as agricultural community at large. However, there are several flaws need to be addressed before the acceptance of the manuscript.

My main comments are:

• The manuscript presented a result of one-year field studies in one location with 5 reps. One of the key features of the dryland cropping system in the SW QLD is high climatic variability (see your statement in line 224 on the extreme weather variability of this region). This study is limited to single year and location and did not capture at least two years of this variability. In addition, while the 5 replications of the experimental treatments could provide an estimate of the variability around the measurements, the authors did not present any measure of error (e.g standard error) around the estimate of the mean (see Fig 3-5, Table 6 for example) of the measured variables. This limits the reproducibility or the relevance of the findings from this manuscript.

• The introduction and the results are very extensive, and the discussion is more general, without going into the effects on the soil that caused a greater or lesser use of N and consequently of the use of water (accumulation of C, nitrification, denitrification, immobilization, etc.), such as for example activity of microorganisms, and their effect on N-mineral transformation in soil? Thus, I believe that some changes should be made to provide a more focused discussion.

• The text in general is very speculative, with a long, repetitive introduction, the material and methods, it needs more information about the herbicide spray, definitions of fertilizers used in the cultures. For e.g what is the composition of the Starter Z (40kg/ha)?

• The authors tried a lot, but the way of personation is still poor. Avoid we, I, you, they, etc. in the manuscript.

• There are extensive citations in the manuscript. In this sense, the manuscript provides a lot of information, but with data that do not add much information relevant to the proposed theme. Overall, the article must be improved with more clarity and a set of information with scientific justification.

• The results and discussion are too long and unwieldy. There is need to streamline it based on the hypothesis of the paper. Many of the figures are repetitive.

Some line-by-line recommendations

L17-18: The practice cover cropping has a limited application to some part of the globe (e.g arid areas or even some areas with unimodal rainfall pattern). May be change to “many part of the globe”

L20: change objectives to ‘objective”

L21: change “different terminations of summer cover crop practices” to “different timing of summer sorghum cover crop termination”

L22: I didn’t see results of interaction. Perhaps it was a mediating effect

L23: remove “planted after a summer sorghum cover crop”

L28-29: “Under late terminated summer cover crop, soil water depleted by 7% at wheat 29 planting which resulted in 61% decline in yield” to ‘Under late terminated summer cover crop, there was 7% soil water depletion at wheat planting which resulted in 61% decline in yield”

L37: soil water retention is different from soil water storage. Paraphrase.

L66: “for increased water used efficiency” or “for increased water use efficiency”

L66: I think the contention that cover cropping has been adopted across the globe is not factually true.

L143: Check GRSP

L164-168: Extensive studies- only 1 cited reference

L179: check “[80] reviewed that an increase”

L226: “2%” hanging

L279: 5.0 cm height of …

L280: remove “height”

L290-294: Describe what was done not what ought to be done

L332: What does the phrase “soil microbial health” means? Perhaps “soil health”

L412: fallow (control) or control… use a consistent phrase for the control treatment.

L420-421: This did not make any sense “There was little difference between control and early termination difference varied across the profile (Figure 3)”

L440-441: This sentence is not necessary” Following the summer cover crop, wheat was planted in plots that were under previous 441 treatments, here we call plots with their treatment name during summer”

L593: The phrase “microbial health” did not make any sense

L952-608: This repetition of the previous section

Reviewer #3: suggestions:

Line 28: ...soil water was depleted by ....

Line 66: ... for increased water use efficiency ...

Line 119: ..., which can be 58-60% of SOM. (delete OC)

Line 239: three months ......

Line 239: Thus, the examined ....

Line 294: is it 20 or 40 tubes?

Line 314: subsamples per plot or per subplot? same for lines 335 and 336.

Lines 552-553: ... contents (TOC, POC and POXC) and clay content (Figure 6).

Lines 579-580: Results also showed that the surface layer 0-15cm stored soil water at wheat ...

Lines 592-593: ... ground cover, organic matter, stored soil water, soil quality, and health.

Lines 595-596: ANOVA test showed the ....

Lines 598-600: ... and increased yield by compared to ...?

Line 600: In this treatment TOC and POC changed by 7% ... (changed? increased or decreased?)

Line 601: ... soil biology as evidenced by an increase ...

Line 603 ... growing season of winter crop showed that managing summer cover ...

Line 604 ... during sensitive phenology stages ... (spelling)

Line 616 ... while enhancing chemical and biological ...

Line 624: ... soil water content at wheat sowing with the control, but enhancements ...

Line 625: ... the increase in yield and grain quality that was observed.

Line 641: However, mid and late-terminated plots ...

Line 688: delete 'could'

Line 724: spelling "concentration'

Line 753: delete 'had'

Line 758: ... affected soil water availability at wheat sowing time.

Lines 761-763: re-write for clarity.

6. PLOS authors have the option to publish the peer review history of their article (what does this mean?). If published, this will include your full peer review and any attached files.

Reviewer #1: **Yes: **Mukhtar Ahmed

Reviewer #2: **Yes: **Ismail Ibrahim Garba

Reviewer #3: **Yes: **Partson Mubvumba, Ph.D.

---

## [Author Response · Author response to Decision Letter 0]

21 Dec 2022

Dear Editor and Reviewers, 

All comments for the manuscript (PONE-D-22-23600) have been addressed. Please see "Response to Reviewers" Document for all revised changes. Much appreciated for your time and recommendations.

Wish you have a Merry Christmas and a Happy New Year!

Kind regards,

Hanlu Zhang

---

## [Decision Letter · Decision Letter 1]

14 Mar 2023

PONE-D-22-23600R1Cover cropping impacts on soil water and carbon in dryland cropping systemPLOS ONE

Dear Dr. Zhang,

Thank you for submitting your manuscript to PLOS ONE. After careful consideration, we feel that it has merit but does not fully meet PLOS ONE’s publication criteria as it currently stands. Therefore, we invite you to submit a revised version of the manuscript that addresses the points raised during the review process.

We look forward to receiving your revised manuscript.

Kind regards,

Muhammad Riaz

Academic Editor

PLOS ONE

Journal Requirements:

Additional Editor Comments:

Your revised submission has been assessed. Although the manuscript has been improved during the revision, there are still some issues which need to be addressed as highlighted by our reviewer(s).

Reviewers' comments:

Reviewer's Responses to Questions

**Comments to the Author**

1. If the authors have adequately addressed your comments raised in a previous round of review and you feel that this manuscript is now acceptable for publication, you may indicate that here to bypass the “Comments to the Author” section, enter your conflict of interest statement in the “Confidential to Editor” section, and submit your "Accept" recommendation.

Reviewer #2: (No Response)

Reviewer #3: All comments have been addressed

2. Is the manuscript technically sound, and do the data support the conclusions?

Reviewer #2: Yes

Reviewer #3: Yes

3. Has the statistical analysis been performed appropriately and rigorously? 

Reviewer #2: Yes

Reviewer #3: Yes

4. Have the authors made all data underlying the findings in their manuscript fully available?

Reviewer #2: Yes

Reviewer #3: Yes

5. Is the manuscript presented in an intelligible fashion and written in standard English?

Reviewer #2: Yes

Reviewer #3: (No Response)

6. Review Comments to the Author

Reviewer #2: The author has addressed most of the comments raised. Overall, I commended the authors for addressing the key issues. However, I still think the text is too long particularly the introduction section. The authors indicated that because they have looked at different aspects of the cover crop effects, the text has to be this long. This is research manuscript not a review article. However, the same information can be presented with shorter introduction of 5-6 paragraphs. The introduction reported of several long-term cover crop studies on soil water, SOC and indicators of soil health, and readers might think this study will also be a long-term experiment. I recommend the introduction to be shorten, by providing the background on the study, the existing knowledge gap(s) on the subject and the need for the reported study and finally a brief summary of the work and its significance. A total of 94 citations was included in the introduction, and as such the text is unwieldy.

Other minor line by line recommendations:

24: Change "on-farm trials" to "on-farm trial" as it is only one trial.

30: late termination or "late terminated cover crop" Be consistent as you used "early/late terminated" before.

86: reduce rather than "prevent" soil water loss from evapotranspiration. The cover crop must use water even if terminated early.

104: Soil organic material or soil organic matter?

236: Trial site not sites

386-514: Should how the % relative change be described in the data analysis sub-section?

448-449: soil water became similar to other treatments at the flowering stage... How soil become similar to other treatments? Perhaps not significantly different?

567: Incorporation of cover crops into what?

569-583: This is already presented results not discussion. Can be remove.

614-619: did the residue affect wheat emergence in the current study? No results on establishment count?

726: POC or POXC?

Reviewer #3: This study shows well how cover crop terminating timing plays an important role in managing soil water storage and yields under dryland cropping systems. The study also tracks how chemical and biological indicators (TOC, POC, AMF sequence concentration, and microbial activities) tie up together in enhancing soil health. This study is particularly important in that it highlights the importance of cover crops under dryland conditions in semi-arid areas, where generally farmers are skeptical about cover crop adoption based on the idea of saving moisture for the main crop. Well-managed cover crops in semi-arid dryland are potentially more beneficial than elsewhere where precipitation is in abundance.

7. PLOS authors have the option to publish the peer review history of their article (what does this mean?). If published, this will include your full peer review and any attached files.

Reviewer #2: **Yes: **Ismail Ibrahim Garba

Reviewer #3: **Yes: **Partson Mubvumba

---

## [Author Response · Author response to Decision Letter 1]

4 May 2023

Thank you Editor Muhammad, reviewer Ismail and Partson for your time and valuable comments!

---

## [Decision Letter · Decision Letter 2]

23 May 2023

Cover cropping impacts on soil water and carbon in dryland cropping system

PONE-D-22-23600R2

Dear Dr. Zhang,

We’re pleased to inform you that your manuscript has been judged scientifically suitable for publication and will be formally accepted for publication once it meets all outstanding technical requirements.

Kind regards,

Muhammad Riaz

Academic Editor

PLOS ONE

Additional Editor Comments (optional):

I am pleased to inform you that your revised manuscript is accepted for publication.

Reviewers' comments:

Reviewer's Responses to Questions

**Comments to the Author**

1. If the authors have adequately addressed your comments raised in a previous round of review and you feel that this manuscript is now acceptable for publication, you may indicate that here to bypass the “Comments to the Author” section, enter your conflict of interest statement in the “Confidential to Editor” section, and submit your "Accept" recommendation.

Reviewer #2: All comments have been addressed

2. Is the manuscript technically sound, and do the data support the conclusions?

Reviewer #2: Yes

3. Has the statistical analysis been performed appropriately and rigorously? 

Reviewer #2: Yes

4. Have the authors made all data underlying the findings in their manuscript fully available?

Reviewer #2: Yes

5. Is the manuscript presented in an intelligible fashion and written in standard English?

Reviewer #2: Yes

6. Review Comments to the Author

Reviewer #2: The authors have addressed all the recommendations. The manuscript can be accepted in the revised form.

7. PLOS authors have the option to publish the peer review history of their article (what does this mean?). If published, this will include your full peer review and any attached files.

Reviewer #2: **Yes: **Ismail Ibrahim Garba

---

## [Editor Report · Acceptance letter]

26 May 2023

PONE-D-22-23600R2 

Cover cropping impacts on soil water and carbon in dryland cropping system 

Dear Dr. Zhang:

I'm pleased to inform you that your manuscript has been deemed suitable for publication in PLOS ONE. Congratulations! Your manuscript is now with our production department. 

Kind regards, 

on behalf of

Dr. Muhammad Riaz 

Academic Editor

PLOS ONE